# Non-invasive brain stimulation paradigms in treatment of alcohol use disorder: Systematic review and network meta-analysis protocol

**Anastasia Demina**[1,2]*, **Benjamin Petit**[1], **Benoit Trojak**[1,2]

**1** Addiction Medicine Department, CHU Dijon Bourgogne, Dijon, France, **2** INSERM U1093, CAPS, Université de Bourgogne, UFR STAPS, Dijon, France

* anastasia.demina@chu-dijon.fr

## Abstract

### Background

Alcohol use disorder (AUD) is a chronic condition linked to allostatic neuroadaptations in the brain's reward circuitry, leading to compulsive and automatized alcohol use in response to craving or negative affect. There are only a few treatment options for AUD, and their efficacy and tolerance profiles remain suboptimal. New AUD management strategies are actively being investigated, and among these, non-invasive brain stimulation (NIBS) interventions. We are planning to conduct a systematic review and network meta-analysis to simultaneously compare different NIBS strategies for AUD, and the present protocol aims to document our methodological approaches and a priori decisions.

### Methods and analysis

We will include only randomized controlled trials involving adults with AUD, alcohol dependence, or alcohol abuse. The primary interest outcomes of our review will concern alcohol consumption in AUD population. In trials investigating NIBS as a strategy for alcohol use reduction, we will explore the effect of NIBS on the reduction in total alcohol consumption and the number of heavy drinking days among participants. In trials in recently detoxified AUD patients where the potential of NIBS to prevent relapse is explored, the primary outcome will concern the rate of relapse. Data on craving and safety parameters will be gathered as secondary interest outcomes. At the time of submitting this protocol, four electronic databases (EMBASE, PubMed, PsycINFO, and The Cochrane Library) and three clinical trial registries (Clinical Trials, EU Trials, WHO ICTRP) were searched. The results of the searches were screened in a blinded manner by two authors using titles and abstracts, with conflicts adjudicated by a third author. The second round of selection based on full texts will be performed after the protocol submission. Data will then be extracted independently

**Data availability statement:** No datasets were generated or analysed during the current study.

**Funding:** The author(s) received no specific funding for this work.

**Competing interests:** The authors have declared that no competing interests exist.

**Abbreviations:** AUD, alcohol use disorder; NIBS, non-invasive brain stimulation; NMA, network meta-analysis; rTMS, repetitive transcranial magnetic stimulation; tDCS, transcranial direct current stimulation; DSM, Diagnostic and Statistical Manual of Mental Disorders; MeSH, medical subject headings.

by two authors using a predefined extraction form. Risk of bias evaluation for each trial will be performed independently by two authors using the revised Cochrane risk-of-bias tool for randomized trials (RoB 2). We will quantitatively synthesize the extracted results using mean differences and risk ratios as effect measures. Initially, a random-effects pairwise meta-analysis will be performed to compare treatment and control arms across different trials. A network meta-analysis will then be conducted. The results of the network meta-analysis will be presented as a network graph representing treatment nodes and direct comparisons, a league table with both direct and network meta-analysis (indirect or mixed) estimates, a net heat plot for inconsistency evaluation, and CINeMA evaluation of the confidence in our results.

## Systematic review registration

PROSPERO registration number CRD42024504362.

---

## Background

The global burden of disease and injury attributable to alcohol is significant: worldwide, 2.6 million deaths were directly linked to alcohol use in 2019 [1]. Alcohol use disorder (AUD) is a chronic condition characterized by allostatic neuroadaptations in the brain's reward circuitry, leading to compulsive and habitual alcohol use in response to craving or negative emotional states [2–4]. Current therapeutic options for AUD are limited, with suboptimal efficacy and tolerance profiles. A recent network meta-analysis of available treatments for AUD found no high-level evidence supporting the effectiveness of any medication in enabling controlled drinking [5,6]. As a result, new approaches to managing AUD are being actively explored, including non-invasive brain stimulation (NIBS), which shows therapeutic promise in helping individuals reduce their alcohol consumption [7].

NIBS techniques are often contrasted with deep brain stimulation, which requires surgical procedures to implant electrodes directly into the central nervous system [8]. NIBS is an umbrella term encompassing various neuromodulation paradigms that use non-invasive stimulation methods. These include electrical current delivery, as in transcranial direct current stimulation (tDCS); magnetic impulse delivery, as in repetitive transcranial magnetic stimulation (rTMS) or in theta-burst stimulation; and ultrasound or shock-wave stimulation, as in transcranial ultrasound stimulation or in transcranial pulse stimulation [9–15]. These transcranial techniques are feasible and well-tolerated by patients [14–17]. While electroconvulsive therapy and magnetic seizure therapy are also classified as NIBS, they require general anesthesia, have different tolerance profiles, and are not routinely explored as treatment options for non-comorbid AUD [18,19].

Multiple trials have investigated the efficacy of NIBS in AUD [7]. Some of these studies contributed to the formulation of a Level B (probable efficacy) international recommendation for the use of tDCS in AUD in 2020–2021 [9,20]. However, as the field of neuromodulation for AUD continues to expand, there is a pressing need for

a systematic effort to synthesize all available evidence in the field. Our network meta-analysis (NMA) project aims to address this by conducting a simultaneous comparison of all existing NIBS paradigms for AUD. The NMA methodology allows inference for indirect comparisons and provides mixed estimates of effects which are more precise than the results from direct comparisons [21]. Since the validity of an NMA depends on the soundness of its assumptions and the quality of the included data, we aim to thoroughly document our efforts to ensure robust assumptions, high-quality data selection and extraction methods, and well-defined pre-specified analyses and dissemination plans.

## Methods and analysis

We followed the PRISMA-P guidelines to ensure that all recommended information is included [22,23]. Our NMA was preregistered in the International Prospective Register of Systematic Reviews (PROSPERO) under registration number CRD42024504362.

As our systematic review and network meta-analysis will utilize already available data from the included trials, ethics approval and consent to participate are not applicable to our analysis. Our dissemination plan includes the publication of the systematic review and meta-analysis in an international peer-reviewed journal, as well as the international communication of our results.

### Eligibility criteria

We will include randomized controlled trials involving participants diagnosed with AUD as defined in the Diagnostic and statistical manual of mental disorders (DSM), 5th edition, alcohol dependence or abuse as defined in previous DSM editions, or alcohol dependence syndrome or abuse as defined by International Classification of Diseases [24,25]. All other types of trials will be excluded to avoid the risk of bias associated with non-randomized or non-controlled studies. To ensure the validity of the transitivity assumption, we will exclude studies that exclusively recruit participants with comorbid conditions, such as depression or post-traumatic stress disorder (PTSD). Including trials focused solely on comorbid populations could compromise the transitivity assumption by altering the distribution of disease severity across trials [21].

We will include trials comparing an active NIBS paradigm to a sham intervention, either as a stand-alone treatment or as an add-on therapy. This distinction will be accounted for when defining the nodes of our NMA. All types of NIBS paradigms will be considered, regardless of stimulation parameters such as cortical target, intensity, frequency, number of sessions, or number of pulses delivered. These broad inclusion criteria were defined in the aim of gathering all available evidence to construct a comprehensive network of interventions and determine their comparative effectiveness.

The comparator will consist of sham NIBS interventions. In cases involving combined treatment regimens, sham interventions paired with an active co-intervention will also be included, with this accounted for by splitting nodes in the NMA.

We will include studies in any language, from all geographical regions, with no restrictions on the year of publication. All types of randomized controlled trial reports will be considered, including journal articles, conference abstracts, theses, and book chapters. If necessary, we will contact authors to obtain supplementary information.

### Outcomes

The primary interest outcomes of our review will concern alcohol consumption in AUD populations. In trials investigating NIBS as a strategy for alcohol use reduction, we will explore NIBS effects on the reduction in total alcohol consumption and the number of heavy drinking days among participants, assessed at immediate post-treatment and, if applicable, including the follow-up period. As follow-up durations may vary across trials, we will define short-term follow-up as less than 1 month, mid-term as 2–3 months, and long-term as more than 3 months after the end of treatment. Separate quantitative syntheses will be conducted for each follow-up period. These alcohol reduction outcomes are supported by international recommendations as clinically relevant [26,27]. Total alcohol consumption is defined as mean daily alcohol

consumption in grams of ethanol per day, and heavy drinking day is defined as a day with more than 60 grams of ethanol consumed in men and more than 40 grams in women. In alcohol reduction trials, the mean difference and its standard deviation will be used as the measure of effect. In trials in recently detoxified AUD patients where NIBS potential to prevent relapse is explored, the primary outcome will concern the rate of relapse, and the risk ratio will be used as the measure of effect. Special attention will be given to the definition of relapse used by the authors, as relapse is often heterogeneously defined across trials [28]. The preferred working definition of relapse that we will use in our NMA is a return to the previous pattern of alcohol consumption. If a different definition is used in a given study, we will describe it in detail and contact the authors in an effort to harmonize relapse definitions for inclusion in the NMA. Relapse is expected to be defined according to one of the following criteria: return to the previous pattern of consumption; return to any alcohol consumption; percentage of heavy drinking days, as predefined by the study authors; and problems related to alcohol use. If relapse is defined as a return to any alcohol consumption in the majority of trials, we will contact the authors of studies using different definitions to obtain data for all individuals who resumed any alcohol consumption. Conversely, if only a minority of trials define relapse as any alcohol consumption, we will contact the authors of these studies to harmonize the definitions. If harmonization is not possible, the respective trials will be excluded from the quantitative synthesis.

Our analysis will include all available data, and in cases of missing data, we will contact the study authors for clarification.

Data on alcohol craving will also be investigated as part of secondary interest outcomes. To provide a comprehensive assessment of the interventions, we will also examine data on adherence, adverse effects, and acceptability. Definitions of these outcomes will be extracted from each study, and quantitative data will be summarized when appropriate to better characterize the efficacy and tolerance profiles of the interventions studied.

## Information sources

The following databases will be searched: EMBASE, PubMed, PsycINFO, and The Cochrane Library. To ensure a comprehensive search strategy, we will also examine trial registries (ClinicalTrials.gov, clinicaltrialsregister.eu, WHO ICTRP), conference proceedings, and theses to identify unpublished studies. Additionally, we will screen the reference lists of included trials to identify any potentially eligible studies. All searches will be updated and re-run before the final analysis to ensure the inclusion of the most recent evidence.

## Search strategy

We began our scoping searches in November 2024. Our literature search strategy utilized medical subject headings (MeSH) or their equivalents, as well as relevant text words related to NIBS. A draft Medline search strategy was developed by a member of the research team experienced in systematic review searches (AD). This strategy was pilot-tested and peer-reviewed by other team members to ensure accuracy and comprehensiveness. Cochrane filters for randomized controlled trials were applied where appropriate, but no additional filters were used. The complete search strategies for all databases are provided in the Supplemental file (S1 File).

## Data management and study selection

After applying our comprehensive search strategy, all entries were retrieved and uploaded into the systematic review management tool, Rayyan QCRI [29]. The study eligibility form was created by AD and peer-reviewed by all research team members. It is available as a Supplemental file (S1 Fig). Two authors independently assessed all titles and abstracts using predefined selection criteria. Any disagreements were resolved through discussion with a third author (BP).

At the time of protocol submission, the second round of selection, based on the full texts of the previously retrieved entries, had not yet begun. The full texts will be reassessed independently by two authors after submission. Any

 

disagreements will be resolved through discussion with a third author (BP). We will provide an explanation for all entries excluded at this stage. Additionally, two raters (AD and BT) will independently assess whether the entries should be included in the meta-analysis, with conflicts resolved through discussion with the entire research team.

## Data collection process

Data extraction will be conducted independently by two authors using a data extraction form created by AD in consultation with all team members (Supplemental file, S2 File). This form will be pilot-tested on three randomly selected trials, and an adapted version will be applied to all selected trials. Any disagreements will be resolved through discussion with a third author. Patient characteristics (age, gender, disease severity) and study details (number of arms, blinding, randomization process, descriptions of interventions and controls) will be systematically extracted and presented in a tabulated form to facilitate evaluation of the transitivity assumption. Measures of effect will be extracted in the form of means, standard deviations, and sample sizes for alcohol reduction trials, and as number of participants who relapsed after treatment as well as a total number of participants for relapse prevention trials. Information on adverse events will also be extracted for safety evaluation.

In cases of missing data or the need for additional details, study authors will be contacted by institutional email, with up to three email attempts made over the course of one month. Data will be recorded using an Excel spreadsheet.

For studies with multiple reports, two authors (AD and BT) will independently compare the reports using author names, intervention characteristics, study locations, study dates and duration, and baseline sample sizes, as recommended by the Cochrane Handbook [30]. Only one report from duplicate trials will be included to avoid biases from including the same trial multiple times.

For crossover studies, only pre-crossover data will be used.

## Risk of bias in the individual studies

Two authors (AD and BT), both experienced in risk of bias evaluation, will independently assess each trial at outcome level using the revised Cochrane risk-of-bias tool for randomized trials [31]. They will evaluate the following criteria: bias arising from the randomization process, deviations from intended interventions, incomplete outcome data management, measurement of the outcome, and selection of the reported result. Each domain will be categorized as "high risk," "low risk," or "some concerns" to determine the overall risk. Signaling questions will be used to guide the authors in making their judgments. If their independent evaluations do not align, consensus will be reached through discussion with a third author (BP).

The risk of bias evaluation will be visually represented next to the forest plots for each outcome, as recommended in the Cochrane Handbook [30]. This evaluation is a crucial component in assessing the internal validity of the included studies. It will provide essential information to critically appraise the review's results regarding the true effect of the interventions. Importantly, this process enhances the credibility and transparency of the review by comparing individual studies to their registered protocols. We will discuss the results of the risk of bias evaluation and describe the implications for the validity of the review's overall findings.

## Data synthesis

We will quantitatively synthesize the extracted results using mean differences as the effect measure for alcohol use reduction trials and risk ratios for relapse prevention trials. First, a random-effects pairwise meta-analysis will be performed to compare treatment and control arms across different trials. The inverse variance method and random effects model will be used to account for clinical and methodological differences between studies. Sensitivity analyses will be conducted by comparing the random effects model to the fixed effects model to evaluate small sample bias. To estimate the effect size in alcohol use reduction trials, we will use Cohen's d interpreted as small ($d = 0.2$), medium ($d = 0.5$), and large ($d = 0.8$) [32].

For the NMA, we will use graph-theoretical methods in R with the netmeta package [33]. This approach makes it possible to implement trials with more than two intervention arms. The netmeta package adjusts for the correlation of pairwise comparisons in multi-arm trials by appropriately reducing their weights.

Alcohol reduction trials and relapse prevention trials will be synthesized separately.

Network nodes will represent the technical characteristics of the interventions (tDCS, rTMS, or ultrasound neuromodulation). For tDCS and rTMS, we will further split the nodes by target characteristic (lateral prefrontal, medial prefrontal, other). For rTMS, the nodes will also be based on the pulse frequency (low or high). Sham interventions will serve as the reference group. We will be mindful of the need to split the sham node if the sham interventions are too heterogenous in the included trials, and we will thoroughly justify our choice of reference group.

Network geometry will be described alongside the network graph, analyzing the structure of the treatment comparisons. We will describe the interventions, their connectedness, the number of studies informing each comparison, and the sample sizes for each intervention node. Additionally, we will detail the direct comparisons and highlight the most well-supported ones.

Data on potential effect modifiers (e.g., sample characteristics and study methods) from each individual study will be extensively described to allow for an independent evaluation of the transitivity assumption.

The results of the NMA will be presented as a network graph showing treatment nodes and direct comparisons, a league table with both direct and NMA (indirect or mixed) estimates, and a net heat plot to assess inconsistency.

A sensitivity analysis will be conducted by merging the nodes in our meta-analysis based on the technical definition of each technique (e.g., all tDCS participants merged, all rTMS participants merged, etc.).

Between-study heterogeneity will first be explored using the clinical and methodological characteristics of the trials. For the statistical evaluation, we will use $\tau^2$ as a measure of heterogeneity magnitude. This will be expressed as a prediction interval, which will be compared with confidence intervals for each comparison to assess whether there are no concerns, some concerns, or major concerns related to heterogeneity [34]. To further explore heterogeneity, sensitivity analyses will be conducted by excluding outlier studies, studies with a high risk of bias, or studies with significant clinical or methodological differences.

## Meta-biases

Selective reporting biases will be assessed by comparing the pre-published protocols of the included trials with their published results. If the protocol is unavailable, we will compare the Methods and Results sections of the published trials. To explore publication bias, we will use small study effects analysis as a proxy, analyzing the comparison-adjusted funnel plot and performing Egger's test. The results of each evaluation will be described, interpreted, and critically discussed in the text of our review.

## Overall confidence in the evidence

To evaluate the confidence in our results, we will use the CINeMA framework, which is based on the Grading of Recommendations Assessment, Development and Evaluation (GRADE) approach, with specific adaptations for NMA methodology [34]. CINeMA integrates the assessment of within-study bias by incorporating the conclusions from the risk-of-bias assessments of individual trials into its framework. It also integrates reporting bias, imprecision, indirectness, incoherence, and heterogeneity [35]. Contribution matrices will be presented and interpreted to analyze how each individual study informs the NMA results.

## Discussion

Our NMA protocol aims to provide a transparent methodological framework for the first simultaneous comparison of different neuromodulation technologies in patients with AUD. However, we can already identify several limitations that will be addressed in the Discussion section of our NMA.

First, to ensure the transitivity assumption, we decided not to include studies in which AUD is systematically comorbid with other conditions (e.g., trials including only patients with both AUD and PTSD). NMA methodology relies on the assumption that all patients across included trials could, in theory, be randomized across all interventions, as in a multi-arm randomized controlled trial [21]. Therefore, efforts must be made to ensure that the distribution of potential effect modifiers is sufficiently homogeneous across studies. We hypothesized that the presence of psychiatric comorbidity is an effect modifier; thus, we will exclude trials involving only comorbid populations. However, this methodological decision must be taken into account when interpreting our results, as comorbidity is highly prevalent in clinical populations with AUD. To address this limitation, we will perform a subgroup analysis, if possible. Specifically, if some trials report data from subgroups with comorbidities, we will synthesize these results and examine whether intervention effects differ in these populations.

Second, our initial search strategy did not include Psychological Index Terms in the PsycINFO database. As a final round of searches will be conducted before the publication of results, we will address this limitation by incorporating relevant Psychological Index Terms into our final search strategy.

Third, we anticipate potential challenges related to the heterogeneous definitions of alcohol reduction or relapse. Indeed, alcohol use outcomes and their standard definitions vary across studies, and our predefined outcomes may not align with those used in the original trials. Where necessary, we will contact study authors to request clarifications and, if possible, obtain raw alcohol consumption data to address this issue.

We will discuss any limitations identified during the conduct of the NMA and will formulate recommendations for future research on neuromodulation in AUD. In addition, we plan to hold annual meetings to update the searches and incorporate newly published trials into our NMA. If three or more new RCTs are identified, we will update our NMA publication accordingly.

## Supporting information

**S1 File. Search strategies.**
(DOCX)

**S2 File. Extraction form.**
(DOCX)

**S1 Fig. Decision strategy for trials independent screening.**
(TIF)

## Acknowledgments

The authors wish to thank Suzanne Rankin for proofreading this manuscript.

## Author contributions

**Conceptualization:** Anastasia Demina, Benoit Trojak.

**Writing – original draft:** Anastasia Demina.

**Writing – review & editing:** Benjamin Petit, Benoit Trojak.

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
