## [Decision Letter · Decision Letter 0]

9 Jun 2025

Dear Dr. Demina,

We look forward to receiving your revised manuscript.

Kind regards,

Sandra Carvalho, Ph.D.

Academic Editor

PLOS ONE

Journal Requirements:

2. Your ethics statement should only appear in the Methods section of your manuscript. If your ethics statement is written in any section besides the Methods, please delete it from any other sectio

Reviewers' comments:

Reviewer's Responses to Questions

**Comments to the Author**

1. Does the manuscript provide a valid rationale for the proposed study, with clearly identified and justified research questions?

Reviewer #1: Yes

Reviewer #2: Partly

2. Is the protocol technically sound and planned in a manner that will lead to a meaningful outcome and allow testing the stated hypotheses?

Reviewer #1: Yes

Reviewer #2: Partly

3. Is the methodology feasible and described in sufficient detail to allow the work to be replicable?

Reviewer #1: Yes

Reviewer #2: Yes

4. Have the authors described where all data underlying the findings will be made available when the study is complete?

Reviewer #1: Yes

Reviewer #2: Yes

5. Is the manuscript presented in an intelligible fashion and written in standard English?

Reviewer #1: Yes

Reviewer #2: Yes

You may also provide optional suggestions and comments to authors that they might find helpful in planning their study.

Reviewer #1: This study aimed to conduct a systematic review and network meta-analysis to simultaneously compare different non-invasive brain stimulation strategies for alcohol use disorder. The strength of this study was to conduct analysis using a rigorous method of systematic reviews. However, there were some concerns in this study.

First, the authors had better add the limitations of this study in the discussion.

Second, they had better describe a new limitation in the discussion that they did not use Psychological Index Terms in the PsycINFO search strategy.

Reviewer #2: The protocol is reasonably designed, but it is necessary to refine the decision basis of methodology and strengthen the transparency of statistical analysis and processes to enhance scientificity and reproducibility. The specific opinions are as follows:

1. Inclusion and exclusion criteria for research subjects

The criterion of excluding patients with comorbidities such as depression or post-traumatic stress disorder may lead to a deviation between the research sample and clinical practice, as patients with AUD often have comorbidities in clinical settings.

It is recommended to supplement the scientific basis for excluding patients with comorbidities, or consider setting up subgroup analyses to explore the impact of comorbidities on the results.

2. Description of intervention measures

The inclusion criteria for NIBS intervention parameters (such as stimulation intensity, frequency, and treatment course) are too broad, which may lead to excessive heterogeneity of interventions and affect the comparability of network Meta-analysis.

It is recommended to stratify according to stimulation techniques (such as tDCS, rTMS) and parameters, and set subgroup nodes in the network Meta-analysis.

3. Issues in defining primary outcomes

In the alcohol consumption reduction trial, the primary outcomes are "total alcohol consumption and heavy drinking days after 3-6 months", but the evaluation time points of different studies (such as 3 months vs. 6 months) may lead to data heterogeneity.

In the definition of relapse rate, the operational criteria for "relapse" (such as drinking for 2 consecutive days or cumulative alcohol consumption ≥ a certain threshold) are not clarified.

It is recommended to unify the evaluation time windows of primary outcomes as "3 months" and "6 months" after intervention, and conduct subgroup analyses respectively. Clarify the definition of relapse.

4. Lack of heterogeneity assessment methods

Only "net heat plot for inconsistency evaluation" is mentioned, but how to quantify heterogeneity (such as using I² statistic) is not explained, making it impossible to judge the degree of variation between studies.

The specific implementation methods of "sensitivity analysis" (such as excluding low-quality studies or stratifying by region) are not clarified, resulting in non-reproducible methods.

5. Please supplement the GRADE. GRADE (Grading of Recommendations Assessment, Development and Evaluation) is a widely used systematic approach for grading the quality of evidence and strength of recommendations in clinical guidelines and systematic reviews.

**Do you want your identity to be public for this peer review?** For information about this choice, including consent withdrawal, please see our Privacy Policy

Reviewer #1: **Yes: ** Masahiro Banno, MD, PhD

Reviewer #2: No

---

## [Author Response · Author response to Decision Letter 1]

22 Jul 2025

PONE-D-24-54144

Non-invasive brain stimulation paradigms in treatment of alcohol use disorder: systematic review and network meta-analysis protocol

PLOS ONE

Dear Academic Editor, Dear Reviewers,

Thank you for your careful attention and valuable comments on our manuscript.

Please find our responses below.

General Comments

1. We made the necessary modifications to ensure that the manuscript adheres to PLOS ONE’s style requirements.

2. We placed the ethics statement at the beginning of the Methods section.

3. For image formatting, we unfortunately experienced difficulties connecting to the https://pacev2.apexcovantage.com/ server. How can we ensure that our figure adheres to PLOS ONE’s requirements without access to this software?

Reviewer #1:

1 & 2. We added a Discussion section and addressed the limitation concerning the use of Psychological Index Terms in the PsycINFO search strategy in it (L236-260).

Reviewer #2:

1. We added a scientific rationale for this decision and discussed the limitation in the Discussion section (L240-249).

2. We revised the Methods section to clarify how the nodes of the network meta-analysis (NMA) were defined (L201-206).

3. We provided additional detail on the length of follow-up and the definitions of relapse, and we discussed these points in the Discussion section (L111-114, L120-124).

4. We added a paragraph describing the approach to exploring heterogeneity, including pre-defined sensitivity analyses (L218-223).

5. We added a section on the confidence in the overall evidence and explained the use of the CINeMA framework, which is based on the GRADE approach with specific adaptations for NMA methodology (L230-235).

---

## [Editor Report · Decision Letter 1]

6 Aug 2025

Non-invasive brain stimulation paradigms in treatment of alcohol use disorder: systematic review and network meta-analysis protocol

PLOS ONE

Dear Dr. Demina,

Thank you for submitting your manuscript to PLOS ONE. After careful consideration, we feel that it has merit but does not fully meet PLOS ONE’s publication criteria as it currently stands. Therefore, we invite you to submit a revised version of the manuscript that addresses the points raised during the review process.

The revised protocol is methodologically sound, addresses the reviewers’ concerns appropriately, and adheres to PLOS ONE’s standards for systematic review protocols involving network meta-analyses. Before proceeding to final acceptance, I kindly ask for clarification on a few minor scientific points that could further enhance the transparency and reproducibility of the work. Specifically, while you mention harmonizing relapse definitions across studies, please consider specifying the preferred operational criteria (such as heavy drinking days or cumulative consumption thresholds) that will guide data extraction and harmonization. In addition, it would be helpful to clarify how multi-arm studies will be handled in the network meta-analysis—for example, whether you intend to split shared comparator groups or adjust for correlation between arms using specific statistical methods within the *netmeta* package. Lastly, a brief explanation of how the risk of bias assessments (RoB 2) will be integrated into the CINeMA framework for confidence evaluation would improve methodological transparency. These are minor suggestions aimed at strengthening the clarity and scientific rigor of the protocol. I look forward to receiving your final version.

We look forward to receiving your revised manuscript.

Kind regards,

Sandra Carvalho, Ph.D.

Academic Editor

PLOS ONE

Journal Requirements:

Additional Editor Comments:

Thank you for your careful and thoughtful responses to the reviewers’ comments.

The revised protocol is methodologically sound, addresses the reviewers’ concerns appropriately, and adheres to PLOS ONE’s standards for systematic review protocols involving network meta-analyses. Before proceeding to final acceptance, I kindly ask for clarification on a few minor scientific points that could further enhance the transparency and reproducibility of the work. Specifically, while you mention harmonizing relapse definitions across studies, please consider specifying the preferred operational criteria (such as heavy drinking days or cumulative consumption thresholds) that will guide data extraction and harmonization. In addition, it would be helpful to clarify how multi-arm studies will be handled in the network meta-analysis. For example, whether you intend to split shared comparator groups or adjust for correlation between arms using specific statistical methods within the netmeta package. Lastly, a brief explanation of how the risk of bias assessments (RoB 2) will be integrated into the CINeMA framework for confidence evaluation would improve methodological transparency.

These are minor suggestions aimed at strengthening the clarity and scientific rigor of the protocol.

I look forward to receiving your final version.

---

## [Author Response · Author response to Decision Letter 2]

12 Aug 2025

PONE-D-24-54144

Non-invasive brain stimulation paradigms in treatment of alcohol use disorder: systematic review and network meta-analysis protocol

PLOS ONE

Dear Academic Editor, Dear Reviewers,

We thank you for your positive assessment of our revised manuscript and for the valuable advice to further enhance its methodological transparency and reproducibility. Please find our responses below:

1. We provided additional details on the expected definitions of relapse and the proposed methods for harmonization (L118–125).

2. We included information on how the netmeta package accounts for the correlation of pairwise comparisons in multi-arm studies (L197–198).

3. We briefly explained how the RoB 2 assessments will be incorporated into the CINeMA framework (L231-233).

We sincerely appreciate your continued consideration of our manuscript and the valuable comments that have helped improve it.

Dr Anastasia DEMINA

---

## [Editor Report · Decision Letter 2]

5 Sep 2025

Non-invasive brain stimulation paradigms in treatment of alcohol use disorder: systematic review and network meta-analysis protocol

PONE-D-24-54144R2

Dear Dr. Demina,

We’re pleased to inform you that your manuscript has been judged scientifically suitable for publication and will be formally accepted for publication once it meets all outstanding technical requirements.

Kind regards,

Sandra Carvalho, Ph.D.

Academic Editor

PLOS ONE

Additional Editor Comments (optional):

Dear authors,

Thank you for submitting the revised version of your manuscript “Non-invasive brain stimulation paradigms in treatment of alcohol use disorder: systematic review and network meta-analysis protocol” (PONE-D-24-54144).

I am pleased to inform you that your manuscript is now accepted for publication in PLOS ONE. The revisions have addressed the earlier comments, and the protocol is well structured, transparent, and appropriately registered.

Congratulations on your work, and thank you for choosing PLOS ONE as the venue for your research.

---

## [Editor Report · Acceptance letter]

PONE-D-24-54144R2

PLOS ONE

Dear Dr. Demina,

I'm pleased to inform you that your manuscript has been deemed suitable for publication in PLOS ONE. Congratulations! Your manuscript is now being handed over to our production team.

Kind regards,

on behalf of

Professor Sandra Carvalho

Academic Editor

PLOS ONE